# Commercial Thermal Technologies for Desalination of Water from Renewable Energies: A State of the Art Review

Jhon Jairo Feria-Díaz [1,2,*], María Cristina López-Méndez [1], Juan Pablo Rodríguez-Miranda [3], Luis Carlos Sandoval-Herazo [1] and Felipe Correa-Mahecha [4]

1   División de Estudios Posgrado e Investigación, Tecnológico Nacional de México/Instituto Tecnológico Superior de Misantla, Misantla 93821, Mexico; mclopezm@misantla.tecnm.mx (M.C.L.-M.); lcsandovalh@gmail.com (L.C.S.-H.)
2   Facultad de Ingeniería, Universidad de Sucre, Sincelejo 700001, Colombia
3   Facultad del Medio Ambiente y Recursos Naturales, Universidad Distrital Francisco José de Caldas, Bogotá 11021-110231588, Colombia; jprodriguezm@udistrital.edu.co
4   Facultad de Ingeniería, Fundación Universidad de América, Bogotá 111321, Colombia; felipe.correa@profesores.uamerica.edu.co
*   Correspondence: jhon.feria@gmail.com

**Abstract:** Thermal desalination is yet a reliable technology in the treatment of brackish water and seawater; however, its demanding high energy requirements have lagged it compared to other non-thermal technologies such as reverse osmosis. This review provides an outline of the development and trends of the three most commercially used thermal or phase change technologies worldwide: Multi Effect Distillation (MED), Multi Stage Flash (MSF), and Vapor Compression Distillation (VCD). First, state of water stress suffered by regions with little fresh water availability and existing desalination technologies that could become an alternative solution are shown. The most recent studies published for each commercial thermal technology are presented, focusing on optimizing the desalination process, improving efficiencies, and reducing energy demands. Then, an overview of the use of renewable energy and its potential for integration into both commercial and non-commercial desalination systems is shown. Finally, research trends and their orientation towards hybridization of technologies and use of renewable energies as a relevant alternative to the current problems of brackish water desalination are discussed. This reflective and updated review will help researchers to have a detailed state of the art of the subject and to have a starting point for their research, since current advances and trends on thermal desalination are shown.

**Keywords:** desalination; multi effect distillation; multi stage flash; vapor compression distillation; renewable energies

## 1. Introduction

Water on the planet is apparently abundant; however, most of it is salt water, represented as seawater in a high percentage. Seawater is not suitable for human consumption or for most man-made processes. Furthermore, distribution of fresh water throughout the world is not uniform. In some places, fresh surface or groundwater is abundant in sparsely populated places, such as Scandinavia, Alaska, parts of southern South America, northern Russia and Canada. In contrast, there are densely populated areas with growing industrial areas, located in sites with a low fresh water availability; consequently, subjected to a high degree of water stress according to the relationship between demand for water and amount of water available [1]. As stated by the United Nations, more than two billion people in the world live in countries facing high water stress [2], an aggravated situation if UNESCO's projections for the period from 2017 to 2028 are considered, where it predicts a greater demand for water not only for agriculture, whose consumption is 70% of the demand worldwide, but also for energy production and generation [3]. Similarly, climate

change, world population growth, contamination of fresh water sources, accelerated urbanization in cities and expansion of public service networks also contribute to global water stress [4,5]. The World Water Program (WWP) hast estimated that by 2030 only 60% of water demanded will be available for consumption. Furthermore, the Organization for Economic Cooperation and Development (OECD) predicted that by 2050, availability will be lowered up to 55% and by the end of the century, 40% of the world's population will live areas water stress regions [6].

Using seawater as a source of fresh water supply could be a solution to the increasing global water stress. Nonetheless, intensive energy requirements and prohibitive costs of desalination technologies restrain their massive use in many communities affected by water scarcity, even though having unlimited access to seawater [7,8]. Based on the International Desalination Association (IDA), in 2017, total capacity of all operating desalination plants worldwide was of 92.5 million $m^3$/d [3]; however, the electrical or thermal energy used in the desalination process represents about 50% of the total cost of production [9]. The energy amount required for a desalination process depends on the quality of input water, level of water treatment, treatment technology used by the facility, and the treatment plant capacity [7,10,11]. As a substitute or replacement for electrical energy, desalination systems powered by renewable energies represent a real alternative to reduce operating costs in conventional desalination systems [12,13]. Table 1 shows the energy required to produce 1 $m^3$ of fresh water from distinct types of water sources.

**Table 1.** Energy requirements for different water sources.

| Water Source | Energy (kWh/$m^3$) |
|---|---|
| Seawater | 2.58–8.5 |
| Wastewater reuse | 1.0–2.5 |
| Wastewater treatment | 0.62–0.87 |
| Groundwater | 0.48 |

Source: Reproduced from Refs. [7,14,15].

Generally, water desalination processes can be classified into phase change or thermal processes, and processes without phase change or by membranes [1,4,7,16–20]. The phase or thermal change process involves evaporation of salt water by contact with a heating surface (evaporation surface) leaving the salts in it; then, the fresh water vapor condenses in cooling pipes producing high-pressure water with quality and without salts [21].

The phase change or thermal technologies, available and of great commercial use, are Multi Effect Distillation (MED), Multi Stage Flash Distillation (MSF), Mechanical Vapor Compression (MVC), and Thermal Vapor Compression (TVC) [1,17,22,23]. Similarly, there are technologies that directly use solar radiation as an energy supplier, such as Solar Still (SS), Solar Chimney (SC), and Humidification/Dehumidification (HDH) [24], although they are not currently commercially available on a large scale. Desalination process without phase change consists of the use of membranes or any other element or material to directly separate the dissolved salts in the water, applying high doses of energy or pressure. Membrane techniques include Microfiltration (MF), Ultrafiltration (UF), Nanofiltration (NF), Membrane Bioreactor (MB), Membrane Distillation (MD), Electrodialysis (ED), and Reverse Osmosis (RO) [25]. These are pressure-driven processes to remove particles, bacteria, and salts from water by size exclusion through membranes with different pore sizes [26,27].

A diagram of the different technologies available for desalination of seawater or brackish water is shown in Figure 1.

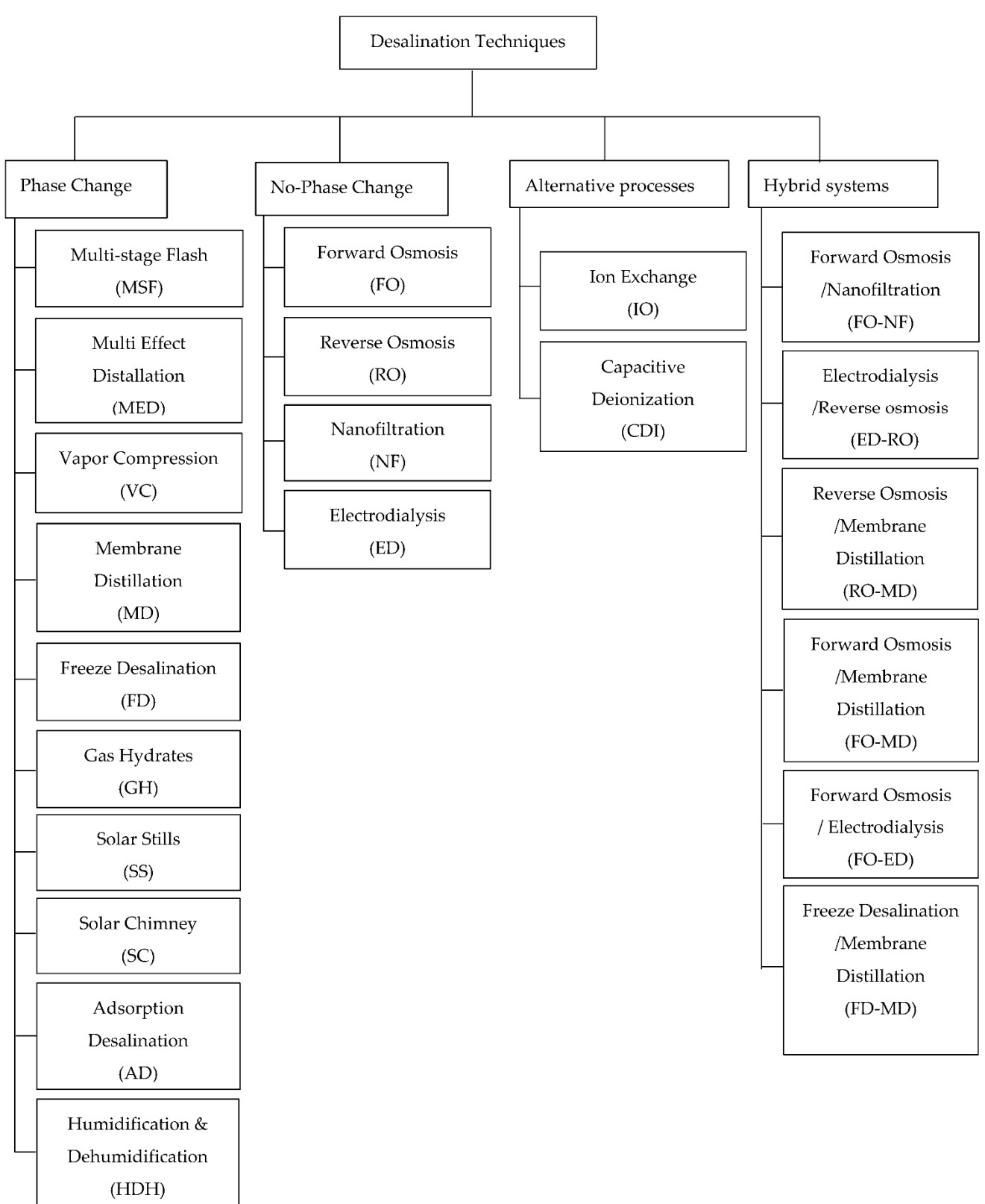

**Figure 1.** Available desalination technologies. Source: Adapted from Refs. [4,17,19,24].

Hybrid technologies for water desalination, such as thermal processes with reverse osmosis, have been developed since the end of the last century [28]. Similarly, the combined benefits of the high separation efficiency of MSF with the low energy consumption of RO

have been highlighted [29]. Nowadays, hybrid desalination technologies cover a broad spectrum, including the integration of RO with other membrane processes, such as ED with MD, and hybridization of RO or MSF with other technologies emerging desalination [4].

This paper aims to present a profound literature review of the different commercial phase-change (thermal) desalination technologies that currently exist and present an overview of the use of renewable energy in water desalination systems and their future perspectives as a contribution to the sustainability of the water resource.

## 2. Phase Change Technologies

The main thermal desalination processes of great commercial use are: MSF, MED and MVC, with a market share of commercial desalination plants of 87.3%, 12.5%, and 0.2%, respectively. Other types of thermal desalination processes such as SS, HDH, and freezing are not found commercially and are limited to experimental prototypes or conceptual designs [30]. The characteristics of the main commercial use thermal desalination techniques and the development of the different optimization strategies for each of these technologies are shown below.

### 2.1. Multi Effect Distillation (MED)

The MED process was the first thermal process implemented in desalination of seawater for consumption. Small MED plants with capacities less than 500 m$^3$/day were introduced to the desalination industry in the 1960s [30,31]. The MED systems have a series of stages or phases, with a decreasing pressure gradient. A heat source is used to increase the temperature of the input water up to 110 °C for the first phase. This heat can be initiated from a boiler running on fossil fuels, waste heat, or renewable resources. Steam is generated in a serial pattern and, in the first stage, it is transferred through a tube to subsequent stages to further boil seawater [32]. This is a medium to high capacity desalination method, where the created vapors are condensed, to give the necessary enthalpy of condensation to the seawater that feeds the system [33]. A scheme of the MED desalination process is shown in Figure 2.

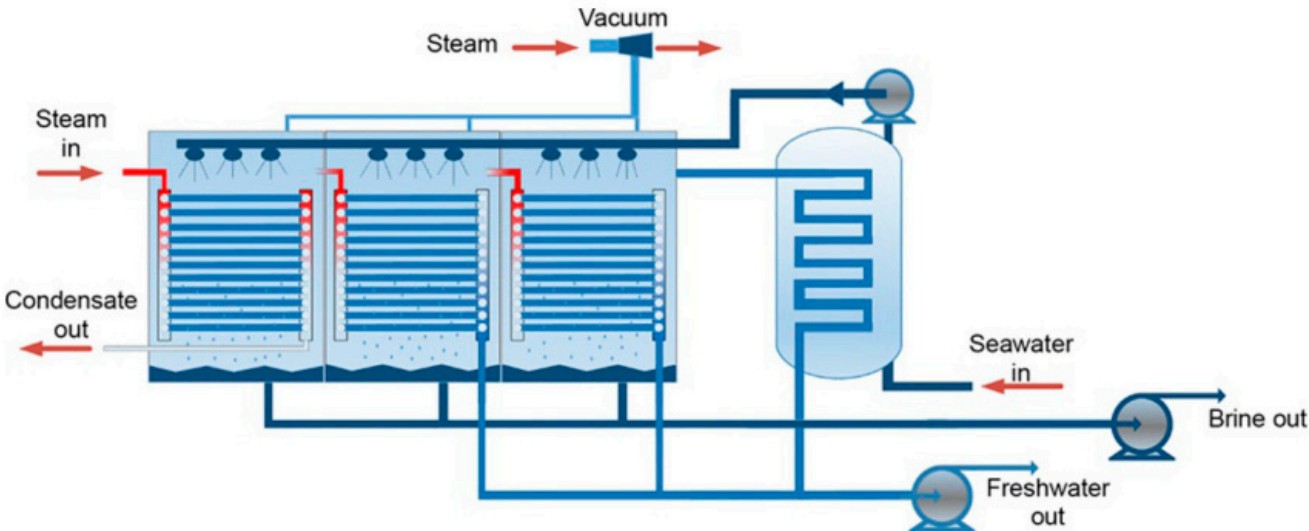

**Figure 2.** Schematic representation of the Multi Effect Distillation process, MED. Source: Reproduced with permission from Ref. [32].

Industrial MED systems include up to 12 evaporation effects, giving them a water production capacity from 600 to 30,000 m$^3$/day. The evaporation in the first phase is driven by the steam extracted from the cogeneration boilers. The steam formed in the first phase is used to drive evaporation in the second effect. This process continues with subsequent phases until the phase temperature drops to around 30–40 °C. Most industrial

MED systems are designed to operate autonomously, where part of the steam formed in the last phase is compressed to the desired temperature and is used to drive evaporation in the first phase [30].

Among the desalination processes, the MED thermal process is a promising one due to its low electrical energy consumption, low operating cost, and high thermal efficiency [11,34]. Based on the energy consumption and heat transfer, MED is more efficient than MSF [21,35,36]. In addition, it has also been shown that MED when combined with other thermal technologies such as MSF and TVC, present higher efficiencies and performance [37]. Similarly, optimization of MED-TVC has been reported when RO is added, achieving greater heat recovery, lower energy costs, lower brine flow and lower salinity in fresh water [38,39]. Hybrid configurations are increasingly promising and efficient than traditional standard thermal desalination configurations.

In the past five years, MED systems research has been focused on five main topics: (1) Simulation with computational models, (2) MED process optimization, (3) waste heat recovery, (4) hybrid systems, and (5) simulations with alternative energies.

On the first topic, steady-state mathematical modeling has been used to simulate parameters that control the MED process [40–43]. These models have been useful to optimize investment and operating costs, and determine the specific value of fresh water based on each MED plant capacity. Theoretical and experimental simulations have also been implemented. Several researchers have simulated different MED configurations [44,45]; in some cases, achieving a reduction of up to 50% in energy and 30% in operating costs compared to a conventional configuration [46]. Among configurations that have aroused great interest, there is the tube-bundle [47] and Boosted MED technology [48], offering greater thermodynamic and economic performance. Normally, results of simulations and economic analyzes showed that decreasing the amount of extraction vapor in MED can significantly reduce the cost of fresh water production [49,50].

On the second topic, MED process optimization has been focused on the preheating of water entering the system and evaporation by spraying. The configurations that implemented seawater preheating increased the performance ratio by up to 10% [51], even recording an average daily performance ratio of 2.5 and an average specific thermal energy consumption of 831 kJ/Kg, using thermal storage tanks and solar collectors [52]. According to the conducted simulations, with the use of the Spray Evaporation Tank, high evaporation efficiencies can be achieved if the required injection/spray parameters, the correct ratio between the water droplet size, and the fall distance are used in conjunction with the temperature of the warm air vapor [53,54]. Similarly, it was shown that the lowest cost of freshwater production is obtained with 17 effects, for certain operating conditions [55]. The third topic shows the viability of the use and recovery of residual heat emitted by industrial furnaces and combustion gases, and its convenience compared to conventional systems [56,57]. Use and recovery of waste heat in MED systems can increase exegetic efficiency by up to 7.34% [58]. Correspondingly, energy recovery through salinity differences in the utilized brine [59], and the use of heat adsorption pumps [60] can serve to optimize the MED system performance.

The fourth topic on hybrid systems with MED shows the development of different MED simulations integrated with other desalination technologies, mainly with TVC. Particularly, convenience of hybrid MED and TVC systems, coupled to power supply systems with solar plants, has been demonstrated. The optimization of this type of systems allowed distillate production to increase by 16.62% and the total exergy to decrease by 3.58% [61–63]. MED and TVC with self-adjusting ejectors have also been simulated to improve the Hybrid system performance [64]. Other desalination systems have been proposed to be coupled with MED, among which are the hybridization of MED with AD [65], MED with Reverse Electrodialysis [66], and MED with MD [67], showing very promising results. Nonetheless, none of these prototypes have been brought to commercial scale and are currently in the research and development stage. Finally, the fifth topic of MED simulations with alternative energies is the one that has caused the most interest among researchers. Several studies

have shown that there are many potential ways to hybridize MED with renewable energies, such as geothermal [68] and concentrated solar energy [69–71]. Theoretical and practical simulations conducted with specialized software in pilot plants, were able to define the optimal criteria for design, optimization, and evaluation of the technical feasibility of future MED systems installations, partially powered by solar energy [72–75]. It has been shown that, in areas with high solar radiation, solar fields can produce much more thermal energy than required by MED units (65 °C minimum) [76], allowing annual production of fresh water to double, if a heating vapor temperature of 90 °C is used instead of 65 °C [77]. Linear Fresnel-type solar collectors have also been used as an alternative for direct supply of solar energy in MED systems [78]. In Qatari operating conditions, 1 m$^2$ of this type of linear collector produces 8.6 m$^3$ of fresh water annually [79,80]. Hybrid MED systems powered by solar energy have shown, under certain operating conditions, to be more efficient than those powered only with electrical energy since the operating costs of desalination plants are reduced [81,82]. On the other hand, optimal design of a thermal storage tank coupled to MED reduces cost of distillate by 19% and increases the capacity factor from 46% to 75% [83,84]. However, only as MED plants powered by solar radiation increase their production capacity, it is possible to reduce production costs associated with the final value of fresh water [85]. Moreover, it has been shown that coupling solar fields to thermal desalination systems and commercial power grids, drastically reduces the environmental impact on the surroundings [86].

*2.2. Multi Stage Flash Distillation (MSF)*

The basic principle of the MSF distillation technique is flash evaporation. The MSF process distills seawater by vaporizing part of the water in various stages under vacuum, arranged in series [87]. In this process, the evaporation of seawater takes place by reducing the pressure rather than increasing the temperature. To get the maximum output and maintain MSF economies, regenerative heating is generally performed. Therefore, this process needs distinct stages for its completion and it is necessary to gradually raise the temperature of the incoming seawater at each stage [88]. In modern MSF plants, multi-stage evaporators in which there are between 19 and 28 stages, are used [89]; although other authors report the number of stages between 4 and 40, which allows the systems of MSF to produce volumes of water in the order of 10,000 to 40,000 m$^3$/day [90]. The operating temperature of the MSF plant is in the range of 90 to 120 °C.

The first MSF plant was built in the 1950s, however, despite the fact that multi stage flash desalination is an energy-intensive distillation process requiring both thermal and electrical energy [32], it was only in 1974 that the Federal Republic of Germany and Mexico developed in Mexican territory, a MSF plant powered by solar energy with a capacity of 10 m$^3$/d with brine recirculation. It had parabolic trough collectors, a double tube flat plate collector, storage tanks and a desalination unit in the plant [90]. MSF's largest desalination plants are in the Persian Gulf. The Saline Water Conversion Corporation's Al-Jubail plant in Saudi Arabia is the largest plant in the world, with a capacity of around 815,120 m$^3$/day [89], while MSF's largest unit located in the United Arab Emirates, is the Shuweihat plant with a capacity of 75,700 m$^3$/day [3].

Among the advantages of the MSF system for seawater desalination, there is the reliability for large-scale production of distilled water, tolerance to the quality of the supply seawater, and the high quality of the distilled water. However, this technology has the disadvantages of high energy consumption and that the plant is heavy and expensive [91]. A schematic of MSF is shown in Figure 3.

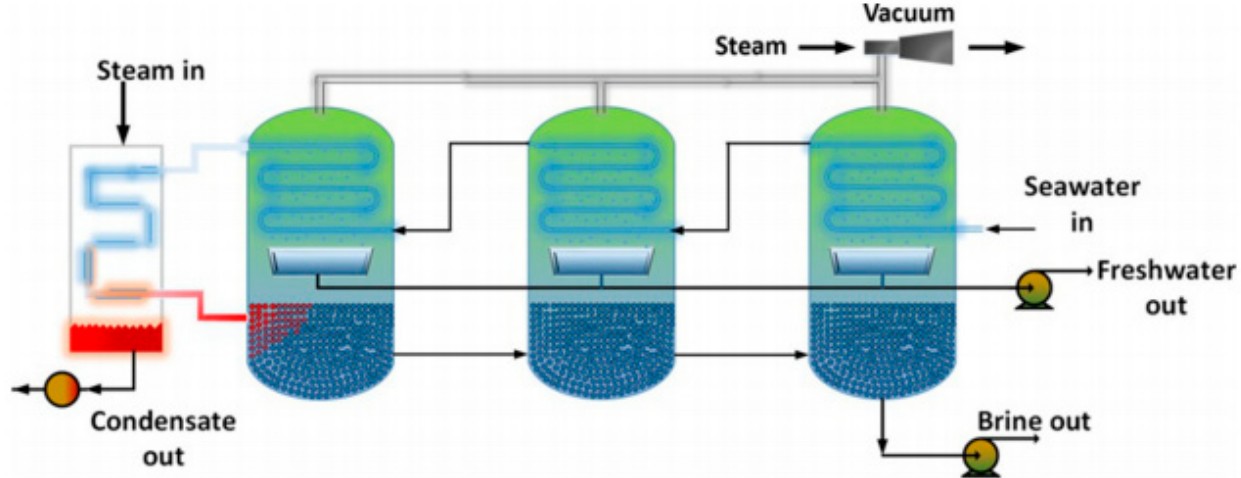

**Figure 3.** Schematic representation of the Multi-Stage Flash Distillation process, MSF. Source: Reproduced with permission from Ref. [32].

With the MSF process, the seawater input is pressurized and heated to the maximum permitted plant temperature. When heated liquid is discharged into a chamber held slightly below the saturation vapor pressure of water, a fraction of its water content is converted to steam. Flash vapor is removed from suspended brine droplets as it passes through a mist eliminator and condenses on the outer surface of the heat transfer pipe. The condensed liquid drips into trays as hot fresh water [92].

Nowadays, several commercial software has been widely used as a modeling and optimization tool for MSF and for other desalination technologies, serving as a basic input in the subsequent design of more complex and larger desalination systems [93]. Hybrid desalination processes based on MSF and RO have also been modeled and optimized, since the combination of these two technologies has greater comparative advantages, such as high general availability, low energy demand and better quality of treated water [94,95].

Research carried out for optimization of MSF units has been focused on exergy and energy savings, process optimization to reduce production costs, and environmental management [96]. Moreover, as in MED for MSF systems, aspects of hybridization and coupling with alternative energies have been research.

By means of exergetic and exergo-economic modeling, fresh water production in MED showed to be significantly higher than MSF (1000 vs. 1521 kg/s) [97]. Nonetheless, it is possible to produce 9000 m$^3$/day of distilled water in MSF plants with 30,000 m$^3$/day of brine, albeit with exergy destruction rates of 71% [98]. In addition, entropy analysis for various MSF configurations have shown that heat transfer is responsible for 65% to 85% of total exergy irreversibility for each MSF stage [99]; however, it is possible to attenuate the consumption of the water specific energy by using condensing steam turbines [100]. On the other hand, an energy analysis carried out at MSF shows that the greatest destruction of exergy occurs in the pumps and motors of the system [101]. Furthermore, it is feasible to reduce exergy destruction in the pumps by more than 39%, in the distillate stream by 29%, and in the purge by 30%, based on optimal MSF settings [102].

For the optimization of MSF systems, simulations have been conducted with software and with experimental monitoring. Both techniques have established that MSF operation on a large scale in cold regions is cheaper than in warm regions due to energy savings in water pumping; fact that should be considered in future large-scale desalination plants [103]. Simulation models with software include simultaneous solutions of mass, moment, and energy; phase equilibria; and seawater properties as a function of temperature, pressure, and salinity [104]; even though, vapor temperature is the only factor that has a significant and positive effect on the distillate flow rate and production-profit ratio [105].

MSF optimization aims to solve two clearly identified MSF problems: required heat supply, especially in remote areas, and high feedwater rejection rate [106]. Recent studies reported that, by reducing the atmospheric pressure in one of the instantaneous vacuum chambers by 20%, the distillation-evaporation ratio improved by 53% and the specific energy consumption was reduced by 35% [107]. Similarly, use of deflectors, special pipes, and sprayers has been proposed in MSF optimization. Simulations that used vertical deflectors and/or changed the deflector angle showed a significant increase in MSF performance [108]. Amount of produced fresh water can be increased by using improved tubes instead of conventional smooth tubes and sprayers to increase the flash evaporation rate [109,110]. Another optimization strategy for MSF consists of brine recirculation [111–116]. With it, MSF can increase the performance coefficient up to 4.4 [117]. On the other hand, incorporation of TVC into MSF in large-scale systems has been simulated with satisfactory results in the performance ratio of this hybrid system [118,119] whereas, at the prototype scale, the development of hybrid systems MSF with FO has proven to be desirable if FO is configured as the system feedwater pretreatment [120,121].

Finally, MSF systems, powered by solar energy to produce electricity and fresh water, have also been thermodynamically modeled, using energy and exergetic approaches [122,123]. The use of parabolic trough collectors (PTC), with an area of 3160 m$^2$, can provide approximately 76% of the energy requirements demanded by an MSF system [124]. Solar energy use by means of PTC makes possible to generate enough energy to achieve high volumes of fresh water in installed MSF plants, with a value of up to USD $2.72 per cubic meter of produced water [125,126], representing an immense potential of alternative energies in optimizing and reducing the operating costs of these thermal desalination systems.

### 2.3. Vapor Compression Distillation (VCD)

VCD is a process used for the evaporation of contaminated saline water, in which the compressed vapors release latent heat. In the vapor compression distillation process, the function of the compressor is to compress the vapors, to increase both their temperature and pressure. Therefore, the latent heat released during the condensation process can be reused to create more vapor [127]. In vapor compression desalination systems, the feed saltwater is heated from the uncondensed vapor, which is mechanically or thermally compressed. The resulting vapor during evaporation overheats due to the increase in the brine boiling point at a pressure lower than the saturation pressure of clean water. If this vapor is compressed to a higher pressure, its temperature increases due to the input of supplementary energy. By increasing its pressure and temperature to the desired level, it can be used as a heat source for the evaporation of brackish water or seawater [33].

Although the principle of compression distillation was known before the 1970s, due to limitation of compressor technology, compression distillation in the seawater desalination field developed slowly. From 1970, with the high-efficiency centrifuge, some difficult problems of the compressors were faced, such as overweight, slow rate, large size, and especially, the compressor shaft sealing technology. Today, the compression distillation technology is quite mature and has become the chosen technology to be combined with other desalination technologies to save energy and reduce costs of the system. For instance, it is possible to obtain high values of the GOR performance index (Gain Output Ratio) in multiple effect stills (MED) when a vapor compressor is coupled to the system, thus improving the unit thermal performance [87,91].

According to the use of devices and energy in the compression process, the compression distillation process is divided into MVC and TVC, where the mechanical compressor works with electricity and the thermal compressor uses an ejector of steam jet to create vacuum [33]. Normally, the production rate of distilled water from MVC (100 to 3000 m$^3$/day) is lower than that of TVC (10,000 to 30,000 m$^3$/day) [32]. Figure 4 shows the schematics of the MVC and TVC processes, respectively.

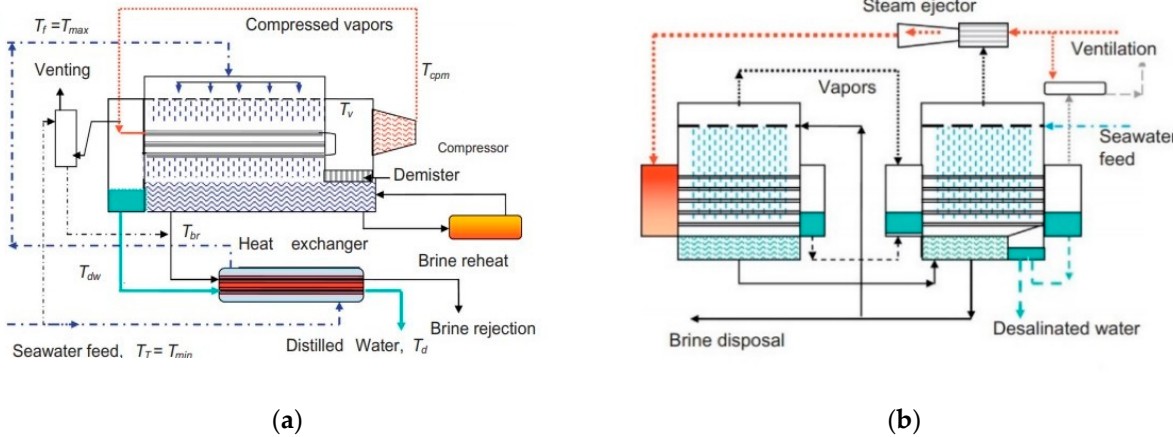

**Figure 4.** Schematic representation of the process: (**a**) MVC and (**b**) TVC. Source: Reproduced with permission from Ref. [33].

Among the advantages that vapor compression distillation (VC) systems have are the high efficiency of converting heat into work, the reduced volume compared to other distillation systems, they do not require large-scale heat sources and that the compressor works with electricity or with diesel engines, so they are suitable for implementation on ships, islands, and maritime bases, i.e., on a reduced scale. Nonetheless, disadvantages are the delicate boiler design and that the scale of water production is limited by the capacity of the compressor [128].

In modern desalination systems with MED, it is common to couple Vapor Compression units, configuring a hybrid treatment system with the ability to reduce specific energy consumption, thermo-economic costs and represent a feasible option in the sustainable management of brine [129,130]. Similarly, hybrid systems have been proposed consisting of compression vapor distillers and humidification and dehumidification units (HDH) to recover the residual heat from the system and improve its efficiency and performance [131].

Most recent publications regarding Vapor Compression Distillation technologies have been focused on thermodynamic and economic modeling aiming to optimize production costs, hybrid systems with thermal and non-thermal technologies, and on VCD systems input with alternative energies. Authors such as Randon and collaborators [132] highlight that MVC system is a very suitable technology for treating high hardness water, economically and efficiently. It must be considered, though, that input parameters used in thermo-economic analyzes have a notable influence on the final value of the cubic meter of distilled water [133]. Nevertheless, it is possible to demonstrate the economic viability of the MVC system to produce fresh water by means of mathematical simulations based on exergy and energy analysis [134]; in addition to achieve the zero emissions targets [135]. On the other hand, TVC systems are coupled to spray-assisted low desalination processes (SLTD) to make these systems economically viable [136,137].

It is worth mentioning the growing interest in the development of hybrid systems that TVC uses to optimize MED systems, due to the high quality of the fresh water produced [138] and the high system performance, if adequate control of operational parameters is carried out [139,140]. In this sense, use and improvements in the design of steam ejectors have shown increases in the performance of MED-TVC systems [141–143]. Self-adjusting ejectors can achieve higher drag ratios [144], and two-stage ejectors can achieve vacuums of approximately 5.3 kPa [145]. Simulation results show that it is possible to improve MED-TVC designs when the compression ratio of the thermal ejector is from 2.1 to 2.6 [146,147]. Exergy and economical simulations have also been performed for hybrid MED-MVC configurations which managed to reduce total fixed production costs by 30% [148].

Hybrid VCD systems with non-thermal technologies are still in the research and development stages. For TVC-RO systems, the simulations indicate that better performances are

obtained when they are configured in series and not in parallel and independent form [149]. Even if the configuration is MED-TVC-Reverse Osmosis, energy loss can be significantly reduced [150]. On the other hand, MVC systems have been proposed as a complement to MD and AD systems. Swaminathan and collaborators [151] coupled MVC with MD and demonstrated that this hybrid system can reduce the final cost of fresh water produced by 6% compared to an independent MVC system. Similarly, Askalany and collaborators [152] proposed a novel MVC-AD system that managed to increase the amount of desalinated water in a range of 10–45% as a function of the conduction temperature of the silica gel used.

VCD systems have also been exergetically and thermodynamically modeled, coupled to solar and wind power systems. The results of the simulations for MED-TVC powered by solar energy show important improvements in the energy performance of the hybrid system compared to when it is powered by conventional electrical energy [153–155]; even when using photovoltaic systems between 35–40% as a renewable photovoltaic contribution in hybrid TVC systems, it is the best combination possible [156].

To summarize, Figure 5 shows the number of publications made on thermal desalination systems in the last five years.

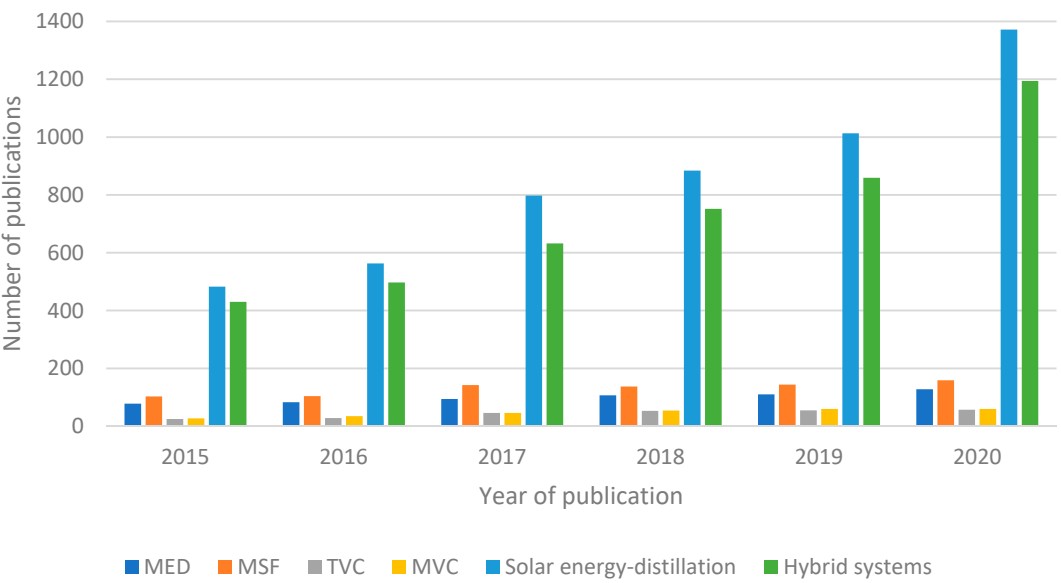

**Figure 5.** Number of publications on thermal desalination technologies (2015–2020).

The scientific research database, ScienceDirect, was used for this technical literature review. A clear and growing interest could be observed in research on the integration of alternative energy and desalination systems, and on the hybridization of these technologies. This tendency is mainly due to the high energy consumption required by thermal desalination systems and their technical limitations, which forces researchers to innovate in the combination of desalination techniques and the use of renewable energies.

## 3. Use of Renewable Energies in Water Desalination

To produce fresh water, conventional desalination systems demand high energy requirements, which are usually obtained from fossil fuels [7,18]. Table 2 shows the specific energy requirements to produce one cubic meter of fresh water from currently available technologies for commercial seawater desalination.

**Table 2.** Typical total electrical energy consumptions in different desalination technologies.

| Desalination Technology | Specific Energy Requirements |
|---|---|
| Multi-Effect Distillation | 14–21 kWh/m$^3$ |
| Multi-Stage Flash | 20–27 kWh/m$^3$ |
| Mechanical Vapor Compression | 7–12 kWh/m$^3$ |
| Thermal Vapor Compression | 16.26 kWh/m$^3$ |
| Seawater Reverse Osmosis | 4–6 kWh/m$^3$ |

Source: Reproduced from Refs. [18,157].

Nevertheless, when comparing the consumption of thermal desalination plants in operation, it is possible to find specific energy requirements and losses. Table 3 shows the comparative results of energy consumption and losses in MED, MSF and MVC systems.

**Table 3.** Requirements and energy losses in thermal desalination systems.

| Desalination Technology | Plant Capacity | Electric Power Consumption | Loss of Electric Power |
|---|---|---|---|
| Multi-Effect Distillation (MED) | 20,000 (m$^3$/d) | 2.5 kWh/m$^3$ | 10 kWh/m$^3$ |
| Multi-Stage Flash (MSF) | 68,333 (m$^3$/d) | 5 kWh/m$^3$ | 10 kWh/m$^3$ |
| Mechanical Vapor Compression (MVC) | 500 (m$^3$/d) | 8 kWh/m$^3$ | - |

Source: Adapted from Ref. [158].

Although the capacity of the plants analyzed is different, MED and MSF showed equal losses of electrical energy, that is, 10 kWh for each m$^3$ of fresh water produced; however, the electrical energy consumption in MSF was double, because in MED the heat transfer is more effective [34]. On the other hand, several options have been developed to improve the energy footprint of desalination technologies, among which is energy recycling and recovery, hybrid processes, process modifications, use of waste heat and integration with renewable energies [21]. Broadly, the efficiency of the low-temperature heat is quantified by the gain output ratio (GOR), which measures the thermal energy consumed in the desalination process, and is defined as the ratio between the mass of distillate and the mass of steam input ($kg_{distillate}/kg_{steam}$). For MED, commercial manufacturers provide a GOR between 10 to 16; for MSF between 8 and 12 and for MVC, a GOR of around 12 $kg_{distillate}/kg_{steam}$ [11]. This explains why the high technological growth trend of MED over MSF [10], in particular, when coupled with other desalination systems to form more efficient hybrid systems, with less environmental impact and higher quality of the water produced.

Generally, thermal-based desalination techniques significantly consume more energy compared to membrane techniques [11,157]. Only 131 desalination plants in the world, representing approximately 1% of the world's water desalination capacity work with energy from renewable sources [10]. However, use of renewable energy for seawater desalination has increased from 2% in 1998 to 19.3% in 2015 [18].

The main conventional renewable energy sources for water desalination are: solar, geothermal, wind, and tidal energy. Solar and wind energy contribute predominantly to the general renewable energy capacity, and to a lesser extent, geothermal and tidal energy [159]. Photovoltaic solar energy represents 43% of the total share of renewable energies in water desalination, while solar thermal energy represents 27%, wind 20% and the remaining 10% corresponds to hybrid renewable energies [10,160]. However, the share of renewable energies is forecast to increase progressively in the near future, albeit in combination with traditional sources, in order to minimize pressure on non-renewable fossil fuels [161]. In

any case, renewable energies can only serve as a complement to other types of energies, due to their limitations and their unpredictable nature [162].

Selecting a suitable solar system requires many considerations, such as locally available solar radiation, plant location, energy storage method, operating temperature range, plant configuration, type of solar collector, fluid power (working fluids), among others. Most solar desalination systems have not been developed as a single system, but are integrations of independently developed components, although some systems require minor changes for better integration [163]. On the other hand, wind turbines represent a mature technology that has been present in many countries for a long time and represents a viable option as a source of energy for desalination systems; however, problems such as public acceptance, correct location of turbines, visual impact, audible noise, interference in communications, and various impacts on the natural habitat and wildlife, have hampered their full development [159]. Use of hybrid renewable solar-wind energy systems has recently been reported with excellent results in isolated islands, and thanks to these integrated generation systems, it is possible to have continuous energy, even if there is no permanent solar radiation [164,165].

Concentrated Solar Power (CSP) plants have gained great interest due to the possible simultaneous cogeneration of water and electricity. The coupling of CSP to MED and RO desalination systems has been the one with the best technical and economic performance; therefore, it is the one with the greatest technological development, although commercially integrated on a small scale [157]. This is mainly because promoting thermal desalination with CSP is not economically feasible on a large scale, since the costs of the production of 1 m$^3$ of fresh water range between US \$0.94 and US \$4.31, mostly affected due to the capital expenditure in the solar field and the operating expense of the desalination plant [166]. Consequently, desalination is a costly process if compared with the USD \$0.53 price of 1 m$^3$ of fresh water when it is produced through a conventional process [74,92]. Although thermal desalination technologies require higher energy demands and have high maintenance costs compared to RO systems [167], both processes are key to fresh water supply. Thus, reducing their energy demands through research avance is equally important for both [7]. As a response, mathematical models and algorithms have emerged to allow simulating the costs of freshwater production with desalination systems, both thermal and membrane, coupled to diverse sources of renewable and non-renewable energy, allowing them to be compared with each other, to optimize decision-making in the selection [168]. Currently, in any case, the only desalination systems coupled with solar energy, with market opportunities and capable of producing up to 20,000 m$^3$/d of fresh water, are based on reverse osmosis driven by parabolic or linear trough concentrators, or with concentrators of plate coupled to micro gas turbines [169].

At the experimental level, MD is among the most promising desalination technologies from the potential use of sustainable energy sources. Fresh water production rates of 3 kg/m$^2$/h have been reported with an electrical and total efficiency of the experimental system of 18% and 71% respectively, based on the use of a concentrated photovoltaic/thermal system [170]. The University of Almerias in Spain proposes the integration of a solar thermal field based on static collectors coupled to a Vacuum Multi-Effect Membrane Distillation unit (V-MEMD), the experimental results show a rate of maximum fresh water production of 5.5 ± 1 L/m$^2$/h [43]. However, the best application of solar-assisted MD for water desalination is for domestic use in single-family homes [171], where the main drawback is the cooling requirements, like any other heating distillation technology. Therefore, membrane distillation systems, when fully developed, will have market opportunities in seawater desalination systems, but with very small production capacities [169].

Another technique that has been presenting advances and improvements for the desalination of saline waters is HDH coupled with different types of renewable energies and cooling and desalination technologies [172–174]. Recent studies reported an increase in the water production rate from 10.8 to 32.1 kg/m$^2$/day thanks to the use of hygroscopic solutions such as kaolin [175]. The design and operation of HDH processes coupled to

solar photovoltaic thermal modules (PVT) for the simultaneous generation of clean water and electricity has been researched; however, the final cost of 1 m$^3$ of water is expensive with this technology, but its high electricity production makes it the cheapest solution for places in critical environmental conditions [176]. In general, it has been shown that the integration of PVT systems with water desalination systems, ensures the polygeneration of products, improves the general efficiency of desalination, and improves the environmental sustainability of these systems [177].

On an experimental scale, another desalination technology that has aroused great interest among researchers is ED coupled to photovoltaic (PV) systems [178] and to hybrid photovoltaic-wind systems [179]. In terms of renewable energy-driven desalination, ED systems are highly valued for their adaptability to varying power conditions, as they can operate at a wide range of direct current voltages [180]. ED systems are more favorable for brackish water desalination with relatively low Total Dissolved Solids (TDS), as it is normally considered to be economically uncompetitive for seawater desalination due to expensive ion exchange membranes, expensive electrodes and relatively short lifespan when working in a high-density electric field [180,181].

In short, the use of renewable energy sources for desalination is essential and decisive if we want to provide an adequate supply of clean water that meets our future needs, reduces harmful impacts on the environment, and is sustainable over time [10]. Consequently, the future of desalination with optimized energy requirements is predicted to include ultra-high permeability membranes, scale resistant membranes, hybrid systems, and renewable energy-driven desalination [7].

Finally, within the strategies to reduce energy waste in thermal desalination plants and take advantage of the maximum useful work possible of the energy during the process (i.e., the exergy), it is possible to propose the integration of these systems with other generation systems of energy, in such a way that a "dual purpose" or "cogeneration of electricity and water" configuration is achieved. Similarly, the integration of thermal desalination processes with renewable energy sources could reduce the massive use of electrical energy from fossil fuels, thus making it more environmentally and economically sustainable [182]. Another strategy aimed at rationalizing energy consumption and taking advantage of exergy in thermal desalination systems consists of the use of system feed water preheaters, in particular MED-TVC. In this way, the energy required to preheat the feed water decreases and will evaporate upon entering the first effect of MED [183].

## 4. Future Perspectives

Despite the evident advances that commercial thermal desalination systems have had in recent decades, which has allowed a considerable increase in the flows of desalinated water produced globally, it is necessary to mitigate the intensive energy requirements to lower the high production costs and make its massive use more accessible in regions in need of this technology and with low per capita income. Present and future research is aimed at significantly improving energy savings and optimizing processes and equipment, to reduce the current limitations of thermal desalination processes. Consequently, the hybridization of technologies and the use of renewable energies is the way to go, because the availability and coupling of emerging energies with hybrid systems is currently more relevant. Considering the limitations of solar radiation as the only energy provider, it is also recommended to hybridize renewable energies to take advantage of their full potential, such as a hybrid solar and wind energy system coupled to water desalination systems.

Nowadays, the race for the next generation of seawater desalination systems has already started with RO and low-temperature MED systems. Its low cost of energy consumption gives it more advantage compared to other systems such as MSF [44]. However, research on emerging desalination technologies have clear merits and environmental benefits over RO, for which the trend is the development of hybrid MED systems with this type of emerging technologies, where the most opted is FO [22]. Additionally, considering MED's low energy consumption, it can be operated with solar energy coupled to a

MED + TVC system. However, it is necessary to deepen research on the configuration of this process, since it is still in the development stage, but with very good prospects on a commercial scale.

On the other hand, use of Waste Heat (WH) as an optimization strategy in conventional desalination technologies has demonstrated its technical viability, since it improves the productions and efficiencies in thermal desalination processes. The use of waste heat in desalination generates significant economic and environmental benefits by eliminating or reducing fuel and energy input. In turn, this will result in the reduction of production costs of desalination and greenhouse gas (GHG) emissions associated with fuel consumption [184]; however, most waste heat driven systems are in a pilot and laboratory scale, with a clear need for further development for large-scale plants. Future planning of a new desalination facility should consider the use of available waste heat sources related to power plants or industrial parks [185].

Regarding the mathematical modeling of processes, energy and exergy analyzes should be used as key factors when applying solar energy systems to improve the energy and exergetic efficiency of the modeled systems [186]. Among the developments and research required to overcome the technological limitations that hinder the massive access of desalination technologies from renewable energies, the following can be mentioned [187]:

- Development of components, intelligent controls, and new materials that allow optimizing the coupling of desalination systems with renewable energy systems.
- Development of new types of desalination membranes that can easily and optimally hybridize with thermal desalination systems such as MED.
- Proper disposal of brine due to the high environmental impacts that its mismanagement represents.
- Deepening the coupling and extensive use of photovoltaic technology as an alternative energy supply in desalination systems.
- Optimization of solid-liquid phase separation processes in the thermal desalination of water.
- Improving the automatic management of energy loading and unloading, as they are necessary for the proper management of the maximum load of solar thermal and photovoltaic energy.
- Development of systems and equipment to improve the conversion efficiency of photovoltaic energy, because the currently available on the market is low.
- Innovation in materials for photovoltaic cells in such a way that they act independently of temperature and local climatic variables.
- It is necessary to develop prototypes and equipment where the desalination process is compact, portable, mobile, and simple in design and manufacturing to produce small-scale water from saline, brackish, and fluoridated water.
- In the thermal desalination system, overlay formation in the pipes and tubes of the heat exchanger is the main issue due to the high-water salinity. It is very difficult to remove overlays and it also reduces the heat exchanger efficiency, therefore, coming up with technical solutions to these difficulties is a need that must be addressed promptly.
- All the limitations and challenges presented can be overcome with innovative ideas and research, which beat knowledge barriers and give development opportunities to experimental research, modeling, and computational simulation. Therefore, collaboration in the field of R&D between academia and industry is essential to transform these future technological developments into commercial products, capable of responding to the needs and problems of society.

## 5. Conclusions

Conditions of water stress that currently exist in many populated areas of the planet, and are expected to increase shortly, have led to the optimization of commercial thermal desalination systems and development of new alternative systems that, although still in the experimental stage and on a pilot scale, represent the best alternative to face this problem.

Optimization of thermal desalination systems has been oriented towards hybridization with commercial (VCD and RO) and non-commercial (HDH, ED, MD, AD) technologies, and the reduction or replacement of electrical and combustion energy by renewable energies, especially by solar and wind, proposing alternation or simultaneity between them. All this, due to the need to reduce the equipment costs used in desalination technologies and the automation required for this type of process.

Recent research in the development of commercial thermal technologies revolves around reducing production costs, reducing impact on the environment and reducing greenhouse gas emissions; therefore, the focus is on increasing desalination efficiencies, recovering heat residual and exergy, mathematically modeling hybridization alternatives, and finally, the best is on integration with renewable energies. In this way and in the future, research will be directed towards the development of desalination technologies that allow to find inexpensive equipment, with low energy consumption and with high performance and efficiency.

**Author Contributions:** Writing—original draft preparation, J.J.F.-D.; writing—review and editing, F.C.-M.; Conceptualization, M.C.L.-M., L.C.S.-H. and J.P.R.-M. All authors have read and agreed to the published version of the manuscript.

**Funding:** This research received no external funding.

**Data Availability Statement:** Not applicable.

**Conflicts of Interest:** The authors declare no conflict of interest.

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
