# Peer review of "Commercial Thermal Technologies for Desalination of Water from Renewable Energies: A State of the Art Review"

_processes, doi:10.3390/pr9020262_

Round 1

Reviewer 1 Report

This problem is directly relevant for journal scope. The manuscript follows the formal regulations of journal. It is important to emphasize that, the manuscript mention a lot of current research work as well. Therefore, the review work is very valuable.

I suggest to accept for publication this work after major revision.

Remarks, suggestions, questions

  1. The chapter 2.1, 2.2 and 2.3 are well transparent and informative. I suggest to mention more hybrid processes.
  2. Please add some works into the literature: doi: 10.1016/j.desal.2015.06.008; doi: 10.1016/j.applthermaleng.2012.08.055; doi: 10.3390/membranes10100265; doi: 10.1016/j.desal.2006.08.020
  3. Please mention some energy rationalization and exergy development solution.
  4. I suggest to add comprehensive analysis about heat requirements of MSF, MED and VCD processes.
  5. Figure 5 is very outdated in a review work. Please add recent data and future forecasts too.

Author Response

Response 1: The following paragraphs were added to chapter 2, following the reviewer's suggestion:

Chapter 2 (Lines 93 to 98)

Hybrid technologies for water desalination, such as thermal processes with reverse osmosis, have been developed since the end of the last century [28]. Similarly, the combined benefits of the high separation efficiency of MSF with the low energy consumption of RO have been highlighted [29]. Nowadays, hybrid desalination technologies cover a broad spectrum, including the integration of RO with other membrane processes, such as Electrodialysis (ED) with membrane distillation (MD), and hybridization of RO or MSF with other technologies emerging desalination [4].

Chapter 2.1 (Lines 136 to 141)

In addition, it has also been shown that MED when combined with other thermal technologies such as MSF and TVC, present higher efficiencies and performance [37]. Similarly, optimization of MED-TVC has been reported when RO is added, achieving greater heat recovery, lower energy costs, lower brine flow and lower salinity in fresh water [38, 39]. Hybrid configurations are increasingly promising and efficient than traditional standard thermal desalination configurations.

Chapter 2.2 (Lines 198 to 203)

Nowadays, several commercial software has been widely used as a modeling and optimization tool for Multi-Stage Flash distillation and for other desalination technologies, serving as a basic input in the subsequent design of more complex and larger desalination systems [93]. Hybrid desalination processes based on MSF and RO have also been modeled and optimized, since the combination of these two technologies has greater comparative advantages, such as high general availability, low energy demand and better quality of treated water [94, 95].

Chapter 2.3 (Lines 249 to 254)

In modern desalination systems with MED, it is common to couple Vapor Compression units, configuring a hybrid treatment system with the ability to reduce specific energy consumption, thermo-economic costs and represent a feasible option in the sustainable management of brine [129, 130]. Similarly, hybrid systems have been proposed consisting of compression vapor distillers and humidification and dehumidification units (HDH) to recover the residual heat from the system and improve its efficiency and performance [131].

Response 2: The references suggested by the reviewer were added to the paper.

DOI

Reference Number on paper

10.1016/j.desal.2015.06.008

[182]

10.1016/j.applthermaleng.2012.08.055

[183]

10.3390/membranes10100265

[93]

10.1016/j.desal.2006.08.020

[158]

Response 3: The reviewer's recommendation was accepted and the following information was added (lines 375-384)

Finally, within the strategies to reduce energy waste in thermal desalination plants and take advantage of the maximum useful work possible of the energy during the process (ie the exergy), it is possible to propose the integration of these systems with other generation systems of energy, in such a way that a “dual purpose” or “cogeneration of electricity and water” configuration is achieved. Similarly, the integration of thermal desalination processes with renewable energy sources could reduce the massive use of electrical energy from fossil fuels, thus making it more environmentally and economically sustainable [182]. Another strategy aimed at rationalizing energy consumption and taking advantage of exergy in thermal desalination systems consists of the use of system feed water preheaters, in particular MED-TVC. In this way, the energy required to preheat the feed water decreases and will evaporate upon entering the first effect of MED [183].

Response 4: The reviewer's recommendation was accepted and the following information was added (lines 274-293).

Nevertheless, when comparing the consumption of thermal desalination plants in operation, it is possible to find specific energy requirements and losses. Table 6 shows the comparative results of energy consumption and losses in MED, MSF and MVC systems.

Table 6. Requirements and energy losses in thermal desalination systems. Source: Adapted from Ref. [158] (Al-Sahali, 2007).

Desalination Technology

Plant Capacity

Electric power consumption

Loss of electric power

Multi-Effect Distillation (MED)

20,000 (m3/d)

2.5 kWh/m3

10 kWh/m3

Multi-Stage Flash (MSF)

68,333 (m3/d)

5 kWh/m3

10 kWh/m3

Mechanical Vapor Compression (MVC)

500 (m3/d)

8 kWh/m3

-

Although the capacity of the plants analyzed is different, MED and MSF showed equal losses of electrical energy, that is, 10 kWh for each m3 of fresh water produced; however, the electrical energy consumption in MSF was double, because in MED the heat transfer is more effective [34]. On the other hand, several options have been developed to improve the energy footprint of desalination technologies, among which is energy recycling and recovery, hybrid processes, process modifications, use of waste heat and integration with renewable energies [21]. The efficiency of the low-temperature heat is usually identified by the gain output ratio (GOR), which measures the thermal energy consumed in the desalination process, and is defined as the ratio between the mass of distillate and the mass of steam input (kgdistillate/kgsteam). For MED, commercial manufacturers provide a GOR between 10 to 16; for MSF between 8 and 12 and for MVC, a GOR of around 12 kgdistillate/kgsteam [11]. This explains why the high technological growth trend of MED over MSF [10], in particular, when coupled with other desalination systems to form more efficient hybrid systems, with less environmental impact and higher quality of the water produced.

Response 5: Figure 5 was removed and recent data on the use of renewable energy in desalination systems and future forecasts were added (Lines 301 to 307):

Photovoltaic solar energy represents 43% of the total share of renewable energies in water desalination, while solar thermal energy represents 27%, wind 20% and the remaining 10% corresponds to hybrid renewable energies [10, 160]. However, the share of renewable energies is forecast to increase progressively in the near future, albeit in combination with traditional sources, in order to minimize pressure on non-renewable fossil fuels [161]. In any case, renewable energies can only serve as a complement to other types of energies, due to their limitations and their unpredictable nature [162].

Reviewer 2 Report

The authors focused their attention on thermal technologies for desalination.The topic is timely and the literature review about the state-of-the art is adequate.

However the review appeared as a summary of a huge mole of information. The manuscript lacks in terms of structure and criticism, pro and cons of the different methodologies are not deeply described and the scientific level of the discussion is very poor. 

Author Response

This paper is not intended to be a critical essay on the pro and cons of thermal desalination technologies. The aim of this work was to present a detailed and updated review of technical literature that serves as a starting point for future research on thermal desalination of brackish water. However, the structure of the paper was complemented with the addition of new references with content on hybrid technologies, recent data and future perspectives of the use of renewable energies in thermal desalination, an analysis and discussion on the energy and heat requirements in thermal processes, desalination and energy rationalization solutions and exergy development.

Round 2

Reviewer 1 Report

Thank you very much for your professional responses. I suggest the acceptance of the manuscript in this present form.

Reviewer 2 Report

Reviews consists of an assessment of the previous researches based on the critical  opinions of the experts.The ‘grafting together’ of technical data from various studies without a scientific discussion and a story line make this manuscript unacceptable for publication.